# TRANS-CAPS: TRANSFORMER CAPSULE NETWORKS WITH SELF-ATTENTION ROUTING

## ABSTRACT

Capsule Networks (CapsNets) have shown to be a promising alternative to Convolutional Neural Networks (CNNs) in many computer vision tasks, due to their ability to encode object viewpoint variations. The high computational complexity and numerical instability of iterative routing mechanisms stem from the challenging nature of the part-object encoding process. This hinders CapsNets from being utilized effectively in large-scale image tasks. In this paper, we propose a novel non-iterative routing strategy named self-attention routing (SAR) that computes the agreement between the capsules in one forward pass. SAR accomplishes this by utilizing a learnable inducing mixture of Gaussians (IMoG) to reduce the cost of computing pairwise attention values from quadratic to linear time complexity. Our observations show that our Transformer Capsule Network (Trans-Caps) is better suited for complex image tasks including CIFAR-10/100, Tiny-ImageNet and ImageNet when compared to other prominent CapsNet architectures. We also show that Trans-Caps yields a dramatic improvement over its competitors when presented with novel viewpoints on the SmallNORB dataset, outperforming EM-Caps by $5.77\%$ and $3.25\%$ on the novel azimuth and elevation experiments, respectively. Our observations suggest that our routing mechanism is able to capture complex part-whole relationships which allow Trans-Caps to construct reliable geometrical representations of the objects.

## 1 INTRODUCTION

Convolutional Neural Networks (CNNs) have achieved state-of-the-art performance in many different computer vision tasks (Krizhevsky et al., 2012; He et al., 2016). This is achieved by local connectivity and parameter sharing across spatial locations so that useful local features learned in one receptive field can then be detected across the input feature space. While such a mechanism is sufficient to learn relationships between nearby pixels and to detect the existence of objects of interest, CNNs often fail to detect objects presented in radically new viewpoints due to the complex effects of the viewpoint changes on the pixel intensity values. This limitation forces us to train each CNN with a large number of data points which is computationally expensive.

Capsule Networks (CapsNets) were introduced to explicitly learn a viewpoint invariant representation of the geometry of an object. In CapsNets, each group of neurons (called a "capsule") encodes and represents the visual features of a higher-level object in an instantiation parameter vector or matrix (which we refer to as the pose vector or matrix throughout this paper). The lower-level capsules (which we refer to as part capsules) estimate the pose of the object parts and hierarchically combine them to predict the pose of the whole object in the next layer. The object-part relationship is viewpoint-invariant, meaning that changes in the viewpoint change the pose of parts and objects in a coordinated way. Therefore, regardless of the viewpoint, we can infer the pose of the whole object from its parts using a set of trainable viewpoint-invariant transformation matrices. Capsule routing mechanisms can therefore learn the underlying spatial relationships between parts and objects. This improves the generalization capabilities of the network due to the underlying linear relationship between the viewpoint changes and the pose matrices. In order to route information between capsules, the part capsules vote for the pose of the higher-level capsules (which we refer to as object capsules). A routing-by-agreement mechanism is employed to aggregate votes (which has been traditionally accomplished using a recurrent clustering procedure) effectively computing the contribution of each part to the object pose.

While various proposed iterative routing mechanisms (such as Dynamic (Sabour et al., 2017) and EM (Hinton et al., 2018) routing) have been shown to be effective in the detection of viewpoint variations, their iterative nature increases computational cost. Prior research has additionally shown that these routing mechanisms may fail to properly construct a parse tree between each set of part and object capsules, partly due to the inability of the network to learn routing weights through back-propagation (Peer et al., 2018). This ultimately limits the performance of CapsNets in real-world image classification tasks. Additionally, the correct number of routing iterations serves as an additional data-dependent hyper-parameter that needs to be carefully selected; failing to optimize the number of routing operations can result in increased bias or variance in the model (Hinton et al., 2018). This issue is amplified when training networks with multiple capsule layers.

In this paper, we introduce a novel routing algorithm called *self-attention routing* (SAR), which is inspired by the structural resemblance between CapsNets and Transformer networks (Vaswani et al., 2017). This mechanism eliminates the need for recursive computations by replacing unsupervised routing procedures with a self-attention module, making the use of CapsNets effective in complex and large-scale image classification tasks. Our algorithm also reduces the risk of under and over-fitting associated with selecting a small and large number of routing iterations, respectively. We compare our proposed routing algorithm to two of the most prominent iterative methods, namely dynamic and EM routing, and the recently published non-iterative self-routing mechanism (Hahn et al., 2019). We evaluate performance on several image classification datasets including SVHN, CIFAR-10, CIFAR-100, Tiny-ImageNet, ImageNet, and SmallNORB. Our results show that our model outperforms other baseline CapsNets and achieves better classification performance and convergence speed while requiring significantly fewer trainable parameters, fewer computations (in FLOPs), and less memory. Moreover, our experimental result on the SmallNORB dataset with novel viewpoints shows that the proposed model is significantly more robust to changes in the viewpoint and is able to retain its performance under severe viewpoint shifts. All source code will be made publicly available.

## 2 Related Work

### 2.1 Capsule Networks

CapsNets were originally introduced in Transforming Autoencoders by Hinton et al. (2011); here they pose computer vision tasks as inverse graphics problems to deal with variations in an object's instantiation parameters. This architecture learns to reconstruct an affine-transformed version of the input image, therefore learning to represent each input as a combination of its parts and their respective characteristics. Sabour et al. (2017) introduced capsules with Dynamic Routing (DR-Caps), which allows the network to learn part-whole relationships through an iterative unsupervised clustering procedure. In DR-Caps, capsules output a pose vector whose length (norm or magnitude) implicitly represents the capsule activation. The vector norm should be able to scale depending on the pose values; representing existence with the vector norm can therefore potentially weaken the representation power of any given capsule layer. Hinton et al. (2018) proposed capsules with EM routing (EM-Caps), where capsule activations and pose matrices are segregated to fit the votes from part capsules through a mixture of Gaussians. While powerful, capsule network's routing procedures have several fundamental limitations:: 1) Iterative routing operations are the bottleneck of CapsNets due to their computational complexity, which limits their widespread applicability in complex, large-scale datasets (Zhang et al., 2018; Li et al., 2018). 2) The number of routing iterations are hyper-parameters that need to be carefully tuned to prevent under and over fitting (Hinton et al., 2018). 3) Lin et al. (2018) showed that even after seven iterative routing operations, the entropy of the coupling coefficient was still large, indicating that part capsules pass information to all object capsules. 4) EM-Caps have difficulty converging and have been shown to be numerically unstable, which limits their applicability in complex tasks (Ahmed & Torresani, 2019; Gritzman, 2019).

Several studies have proposed non-iterative methods to replace the traditional iterative routing mechanisms in CapsNets. STAR-CAPS (Ahmed & Torresani, 2019) combines an attention gate with a straight-through estimator to make a binary decision to either connect or disconnect the route between each part and object capsule. Tsai et al. (2020) proposed an inverted dot-product attention routing mechanism (IDPA-Caps) which generates the routing coefficients between capsules; they unroll the iterative routing procedure and perform the iterations concurrently which helps improve

parallelization. While powerful, the number of concurrent iterations is a hyper-parameter that needs to be tuned, and the unrolling process creates a very large network that is memory intensive. Inspired by Mixture-of-Experts, Hahn et al. (2019) introduced a self-routing mechanism. While non-iterative, self-routing attaches stationary routing weights to specific locations which limits its ability to generalize to novel viewpoints. This also increases the required number of trainable parameters, making it impractical for high-dimensional images.

## 2.2 SELF-ATTENTION

Attention operations bias a network to more informative components of the input in order to improve the discriminative capabilities of the model. This operation has been used to tackle the problem of long-range interactions in sequence modeling and has seen great success across the fields of Natural Language Processing (NLP) (Bahdanau et al., 2014), genomics (Zaheer et al., 2020), speech recognition (Chorowski et al., 2015), and computer vision (Hu et al., 2018; Wang et al., 2017). Various attention mechanisms have been used to improve CNNs by allowing the network to capture interactions between elements of the encoded feature space, which is difficult for a stand-alone convolutional operation (Woo et al., 2018; Hu et al., 2018). Transformer based architectures were introduced by Vaswani et al. (2017) and utilize self-attention as the primary mechanism for representation learning. Self-attention employs the standard dot product operation to generate attention coefficients that effectively capture the long-range interactions between inputs and outputs. These architectures have outperformed recurrent neural networks in a wide range of tasks and have become the SOTA for representation learning (Devlin et al., 2018; Radford et al., 2019; Huang et al., 2018). This concept was then expanded to computer vision applications by treating each output pixel as an element in the self-attention operation, thus allowing a CNN to learn global dependencies between receptive fields (Bello et al., 2019). While powerful, this mechanism generates global dependencies for all pixels, making it memory intensive and computationally cumbersome. This issue was later addressed by restricting the scope of each self-attention operation to local patches as opposed to applying self-attention to the global feature space (Hu et al., 2019; Ramachandran et al., 2019). The Set Transformer is an encoder-decoder architecture that utilizes a self-attention mechanism to cluster a group of independent inputs by modeling the interactions among the elements of the set (Lee et al., 2019). Given that the order of the part capsules does not contribute to the understanding of each object capsule, we took inspiration from the Set Transformer to replace the recurrent routing mechanism with a self-attention-based aggregation of the "votes" in a permutation invariant manner.

## 3 TRANSFORMER CAPSULE NETWORK

### 3.1 MODEL ARCHITECTURE OVERVIEW

The Transformer Capsule Network (Trans-Caps) is a capsule-based neural network architecture where each capsule represents an encoded pose matrix. Trans-Caps starts with a convolutional backbone, followed by a sequence of capsule layers. Note that the choice of the convolutional backbone, the number of capsule layers and the number of capsules per layer varies for different sets of experiments. Given an input image $\boldsymbol{X} \in \mathbb{R}^{H \times W \times D}$, the role of the convolutional backbone is to extract a set of features $\boldsymbol{F} \in \mathbb{R}^{H' \times W' \times D'}$ from the input. The backbone can be either a single convolutional layer, a cascade of a few convolutional layers, or a cascade of residual blocks (He et al., 2016). Previous studies have shown improved performance for more complex datasets when using a residual backbone (Tsai et al., 2020; Hahn et al., 2019). We provide a detailed discussion of the various backbone configurations in the Experiments section. The first capsule layer (`PrimaryCaps`) is a convolutional layer, followed by BN applied to the output backbone features $\boldsymbol{F}$. The outputs are then reshaped to form the primary capsule pose elements. All layers following this layer are convolutional capsule layers (`ConvCaps`) with SAR performed between the layers to construct the pose of the object capsules. With our non-iterative SAR mechanism, the pose computation at all stages of the network can be performed sequentially in one forward pass, yielding numerical stability and efficiency. The final capsule layer (`ClassCaps`) has as many capsules as the number of classes and predicts the pose of the objects, $\boldsymbol{P}_j^L$ where $j \in \{1, ..., J\}$. $J$ and $L$ represent the number of object capsules and the total number of layers, respectively. This layer is followed by a linear `Classifier` which is shared across all class capsules and computes the final class logits as

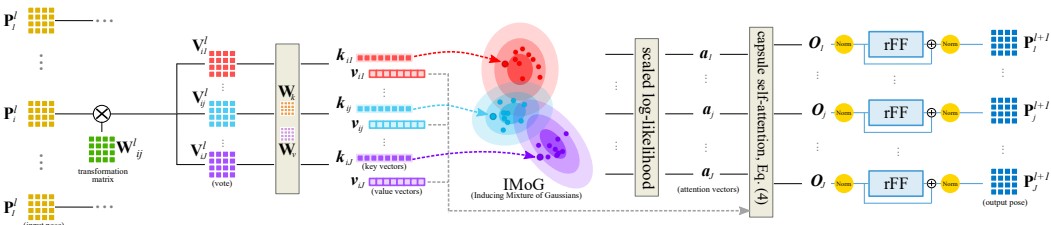

Figure 1: Overview of a Trans-Caps layer with SAR.

$\hat{y}_c = \texttt{Classifier}(\boldsymbol{P}_c^L)$ where $\boldsymbol{P}_c^L$ is the output of the final $\texttt{ConvCaps}$ layer and $c \in \{1, ..., C\}$ in a $C$ class problem.

## 3.2 INPUT POSE TRANSFORMATION

Figure 1 describes one part-object capsule layer with SAR. Let $\Omega_l$ denote the sets of capsules in layer $l \in \{1, ..., L\}$. Each part capsule $i \in \Omega_l$ outputs a pose matrix $\boldsymbol{P}_i^l \in \mathbb{R}^{p \times p}$ in which each of the $p \times p$ entries of the pose matrix represent some instantiation parameter, such as location, size, and orientation. In our implementation, the capsule activation probability is not separately defined, but we instead enable the network to encode the presence or absence of the given part in the pose matrix itself. Each capsule $i \in \Omega_l$ transforms its pose matrix $\boldsymbol{P}_i^l$ and casts a vote $\boldsymbol{V}_{ij}^l \in \mathbb{R}^{p \times p}$ as the prediction made by the $i^{\text{th}}$ part capsule for the pose of the $j^{\text{th}}$ object capsule ($j \in \Omega_{l+1}$). The object-part relationship is viewpoint-invariant and approximately constant; this means that a transformation applied to the pose of a part will have a similar effect on the pose of the object. Therefore, it can be encoded by learned transformation matrices $\boldsymbol{W}_{ij}^l \in \mathbb{R}^{p \times p}$, allowing us to then generate the votes following

$$\boldsymbol{V}_{ij}^l = \boldsymbol{W}_{ij}^l \boldsymbol{P}_i^l \tag{1}$$

Note that the $\boldsymbol{W}_{ij}^l$ transformation matrices are learned discriminatively by back-propagation allowing the network to gradually construct a transformation matrix for each capsule pair to encode the corresponding part-whole relationship. Finally, a non-linear routing procedure is required to process the votes $\boldsymbol{V}_{ij}^l$ for all $i \in \Omega_l$ to then generate the pose of the object capsules in layer $l + 1$. In Trans-CapsNet, we replace the recurrent clustering approach of *iterative* routing methods with our *non-iterative* SAR procedure.

## 3.3 SELF-ATTENTION ROUTING

Our SAR mechanism aggregates votes $\boldsymbol{V}_{ij}^l$ sent to the $j^{\text{th}}$ object capsule according to their agreement and then generates the output that best explains the pose of the higher-level object. Aggregating votes through a self-attention mechanism is sensible as the influence of each vote on the final pose of the object capsule is not necessarily equal. However, if we were to compute the pair-wise similarities among the votes following the self-attention mechanism proposed by Vaswani et al. (2017), we would be limited by the quadratic time complexity of the operation $\mathcal{O}(n^2)$ for $n$ part capsules, making it memory intensive for multiple part capsules. A routing mechanism should also be permutation invariant since the order of the part capsules does not contribute to our understanding of the object capsules. Therefore, we took inspiration from Set Transformers (Lee et al., 2019), in which we generate similarity measures among a set of unordered part capsule votes through interactions with a set of trainable vectors, allowing us to retain linear time complexity $\mathcal{O}(Jn)$ where $n$ and $J$ are the number of part and object capsules, respectively.

We now formally describe our SAR mechanism. We use the following naming convention: $N_h$ corresponds to the number of heads, while $d_k$ and $d_v$ are the respective number of dimensions for the key and value vectors. We also assume that the multi-head attention mechanism evenly divides the $d_k$ and $d_v$ dimensional vectors into $d_k^h$ and $d_v^h$ dimensional key and value vectors per attention head. The operation begins by flattening the vote matrices $\boldsymbol{V}_{ij}^l$ and projecting them onto $N_h$ different

$d_k^h$ and $d_v^h$ dimensional key ($\boldsymbol{k}_{ij}^h$) and value ($\boldsymbol{v}_{ij}^h$) vectors, respectively. This is done using a set of learnable transformation matrices $\Lambda = \{\boldsymbol{W}_k^h, \boldsymbol{W}_v^h\}_{h=1}^{N_h}$, where $\boldsymbol{W}_k^h \in \mathbb{R}^{p^2 \times d_k^h}$ and $\boldsymbol{W}_v^h \in \mathbb{R}^{p^2 \times d_v^h}$. We set $d = d_v^h = d_k^h$ throughout the rest of the paper for brevity, unless otherwise specified. Inspired by the inducing point vectors in Set Transformers, we use an inducing mixture of Gaussian (IMoG) distributions to compute the part capsule agreements. The IMoG is composed of one Gaussian component per object capsule and is parameterized as

$$p(\boldsymbol{x}) = \sum_{j=1}^{J} \phi_j \mathcal{N}(\boldsymbol{x}|\boldsymbol{\mu}_j, \boldsymbol{\Sigma}_j) \quad \text{where} \quad \sum_{j=1}^{J} \phi_j = 1 \tag{2}$$

Note that $\phi_j$, $\boldsymbol{\mu}_j$ and $\boldsymbol{\Sigma}_j$ are the learnable weights, means and covariance matrices associated to the $j^{\text{th}}$ component, respectively. Overall, a trainable IMoG with $J$ components can be seen as $J$ independent memory slots occupied by encoded templates that represent the average appearance of the $J$ corresponding objects. The IMoG templates are *global* query vectors accessed through our attention-based routing procedure to measure the agreement among the part capsule votes. This effectively reduces the quadratic time complexity of key-query self-attention to linear time complexity $\mathcal{O}(Jn)$ (where $J$ is typically small) and allows the self-attention mechanism to properly scale with input size and the number of capsules. Given our use of IMoG, we replace the standard dot-product attention in Transformers with our log-likelihood attention mechanism, which utilizes Gaussians to encode the second-order interactions among points. Assuming a mixture of isotropic Gaussians with $\boldsymbol{\Sigma}_j = \text{diag}[\sigma_{1,j}^2, ..., \sigma_{d,j}^2]$, the similarity matrix $\boldsymbol{S} \in \mathbb{R}^{I \times J}$ from $I$ part capsules to $J$ object capsules computes the log-likelihood of the key vectors $\boldsymbol{k}_{ij}^h \in \mathbb{R}^d$ with respect to the components of the IMoG as

$$s_{ij}^h = \log p(\boldsymbol{k}_{ij}^h; \phi_j, \boldsymbol{\mu}_j, \boldsymbol{\Sigma}_j) = \log \phi_j - 0.5 \log(2\pi) - \sum_d [\log(\sigma_{j,d}^2) + \frac{(\boldsymbol{k}_{ij}^{h,d} - \mu_j^d)^2}{2\sigma_{d,j}^2}] \tag{3}$$

We consider the vector $\boldsymbol{s}_j = [s_{ij}]_{i=1}^I \in \mathbb{R}^I$ as the similarity vector associated to the $j^{\text{th}}$ object capsule. The output of the SAR mechanism for the $j^{\text{th}}$ object capsule from a single head $h$ can then be formulated as

$$\boldsymbol{o}_j^h = \boldsymbol{\mu}_j^h + \sum_{i=1}^{I} a_{ij}^h . \boldsymbol{v}_{ij}^h \quad \text{where} \quad \boldsymbol{a}_j^h = \texttt{softmax}(\boldsymbol{s}_j^h / \sqrt{d}) \tag{4}$$

The similarity vectors $\boldsymbol{s}_j^h$ are scaled by $1/\sqrt{d}$ to avoid the vanishing gradient problem (as was discussed by Vaswani et al. (2017)), and then $\texttt{softmax}$ normalized to sum to one over all part capsules, yielding the vector of attention coefficients $\boldsymbol{a}_j^h$. We emphasize that the $\boldsymbol{o}_j^h$ outputs are the sum of a *static* component $\boldsymbol{\mu}_j^h$, which represents the typical appearance of object $j$, and a *dynamic* component $\sum_{i=1}^{I} a_{ij}^h . \boldsymbol{v}_{ij}^h$. While the static component is input independent, the dynamic component is a function of the input image, and therefore allows the output to account for deformations and variations in the objects' shape and appearance. The raw output is a linear transformation of the concatenation of all attention head outputs given by

$$\boldsymbol{O}_j = \texttt{Norm}[\texttt{concat}(\boldsymbol{o}_j^1, ..., \boldsymbol{o}_j^{N_h})\boldsymbol{W}_o] \tag{5}$$

where $\texttt{Norm}$ can be either a Batch-Normalization (BN) (Ioffe & Szegedy, 2015) or Layer Normalization (LN) (Ba et al., 2016). The final pose of the $j^{\text{th}}$ object capsule is computed as

$$\boldsymbol{P}_j^{l+1} = \texttt{Norm}[\boldsymbol{O}_j + \texttt{rFF}(\boldsymbol{O}_j)] \tag{6}$$

where rFF is any feed-forward layer with ReLU activation that processes each instance independently and identically. As for the $\texttt{Norm}$ function, we selected BN as we found it to be empirically superior to LN in improving the routing operation's convergence speed. BN also enables us to train

Table 1: Key differences between Trans-Caps and the most prominent CapsNets.

| | **DR-Cap** (Sabour et al., 2017) | **EM-Caps** (Hinton et al., 2018) | **SR-Caps** (Hahn et al., 2019) | **Trans-Caps (ours)** |
|---|---|---|---|---|
| **Loss Function** | Margin | Spread | Negative Log-Likelihood | Cross-Entropy |
| **Pose, Activation** | Vector, Scalar (Length of Pose) | Matrix, Scalar | Matrix, Scalar | Matrix, N/A |
| **Routing** | Iterative | Iterative | Non-Iterative | Non-Iterative |
| **Agreement Measure** | Cosine Distance (Dynamic) | Euclidean Distance (Dynamic) | Learned weights (Static) | Log-likelihood Attention (Dynamic) |
| **Normalization & Non-Linearity** | Squash | Sigmoid (for Activation) | BN (for Pose) Sigmoid (for Activation) | BN (for Pose) |

our model with the same optimizer and learning rate as the baseline CNN and CapsNet models, which improves the quality of our comparisons.

## 4 EXPERIMENTS

In our experiments, we evaluate the performance of the proposed SAR mechanism and compare it to baseline CNNs, CapsNets with the four most prominent iterative approaches (Dynamic routing (Sabour et al., 2017), EM routing (Sabour et al., 2017), IDPA-Caps (Tsai et al., 2020), and Variational Bayes Capsule Routing (VB-Caps) (Ribeiro et al., 2020)), and a CapsNet with a non-iterative self-routing mechanism (Hahn et al., 2019). Note that in all experiments, we use the same backbone CNN for all CapsNets for a fair comparison. We finally evaluate the generalization and robustness of the proposed model to viewpoint changes following the key motivation of CapsNets. Our full source-code is publicly available at ***.

### 4.1 TRAINING SETTINGS

The model parameters are updated using a Stochastic Gradient Descent optimizer with a learning rate of $0.1$. In each experiment and for all models, we decay the learning rate by a factor of 10 after a set number of training epochs. We experimented with Spread (Hinton et al., 2018), Margin (Sabour et al., 2017) and cross-entropy loss functions and achieved the best performance using cross-entropy for multi-class classification. Table 1 summarizes the key differences of our proposed model when compared to various CapsNet architectures.

### 4.2 DATASETS

We evaluate our proposed model on various image classification tasks (CIFAR-10 (Krizhevsky et al., 2009), SVHN (Netzer et al., 2011), CIFAR-100 (Krizhevsky et al., 2009), SmallNORB (LeCun et al., 2004), Tiny-ImageNet, and ImageNet (Deng et al., 2009)). The Tiny-ImageNet dataset is composed of a subset of 200 object classes of the original 1000 ImageNet dataset classes, containing 500 and 50 images for training and evaluation, respectively. These datasets of natural images were selected to evaluate the performance of our proposed architecture on complex data tasks. Following Hinton et al. (2018), we select the SmallNORB dataset to evaluate the generalization performance of our model to viewpoint changes. Note that in all the experiments, 10 percent of the training data is sampled randomly to generate the validation set. We perform 5-fold cross-validation and report the average (±std.) error rate over the trained models. Data pre-processing and augmentation details are provided in section appendix A.1.

### 4.3 IMAGE CLASSIFICATION RESULTS

To ensure that our results are consistent with the literature, we follow the CapsNets configurations used in prominent publications (Hinton et al., 2018; Hahn et al., 2019; Tsai et al., 2020). Model architectures for all datasets and their respective output sizes are described in detail in appendix A.4. Figure 2 illustrates the convergence speed of all four CapsNets architectures on CIFAR-10 and the computational cost associated with each model on both CIFAR-10 and CIFAR-100. Trans-Caps has significantly lower memory requirements and has a lower computational overhead (measured in FLOPs) than all other evaluated architectures. We also note that Trans-Caps has nearly the same number of trainable parameters as EM-Caps. We report the error rate of our model across each dataset of interest and compare this result with baseline CapsNets and CNNs in Table 2. The extent

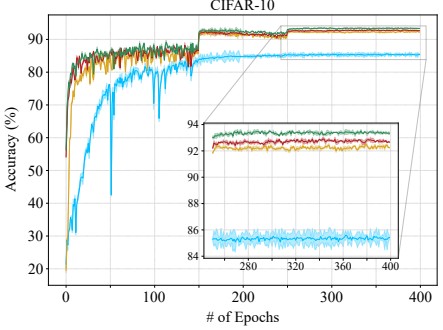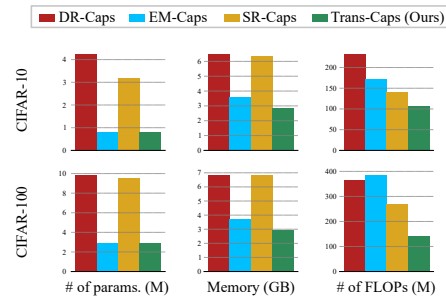

Figure 2: Performance analysis for CapsNets on CIFAR-10 and CIFAR-100. **Left:** Convergence plots for different routing mechanisms on CIFAR-10 validation sets. **Right:** number of trainable parameters, memory usage, and number of FLOPs on CIFAR-10 (top) and CIFAR-100 (bottom). For fair comparison, performance is benchmarked using the same GPU with a batch size of 16.

of improvement between Trans-Caps and the other CapsNets architectures increases with the complexity and variability of the dataset. We note that DR-Caps and EM-Caps degrade substantially as complexity increases, performing poorly on CIFAR-100 and failing to converge on Tiny-ImageNet. SR-Caps performs similarly to Trans-Caps on SVHN but fails to match its performance across the other datasets, and is vastly outperformed on Tiny-ImageNet (20.4% and 36.1% higher top-1 and top-5 error, respectively). Finally, we evaluated the performance of Trans-Caps on full-scale ImageNet; the model architecture is described in detail in appendix A.4. We achieved a 65.13% and 86.81% top-1 and top-5 classification accuracy, respectively.

Table 2: The test error rates between different CapsNet and CNN models on various tasks.

| | SVHN | CIFAR-10 | CIFAR-100 | Tiny-ImageNet top-1 | top-5 |
|---|---|---|---|---|---|
| ResNet-20 (He et al., 2016) | 3.58±0.11 | 8.28±0.24 | 29.10±0.21 | 39.81±0.32 | 17.47±0.27 |
| ResNet-32 (He et al., 2016) | 3.25±0.16 | 7.51±0.18 | 29.86±0.17 | 38.68±0.30 | 16.83±0.26 |
| SE-ResNet (Hu et al., 2018) | 2.85±0.17 | 6.32±0.20 | 24.83±0.26 | - | - |
| AA-ResNet (Bello et al., 2019) | 2.68±0.11 | 4.76±0.13 | 21.43±0.14 | 32.48±0.27 | 13.66±0.23 |
| DR-Caps (Sabour et al., 2017) | 3.44±0.28 | 7.28±0.15 | 47.89±0.42 | - | - |
| EM-Caps (Hinton et al., 2018) | 4.15±0.37 | 14.63±0.43 | 59.03±0.58 | - | - |
| VB-Caps (Ribeiro et al., 2020) | 3.81±0.21 | 11.48±0.22 | 37.28±0.44 | - | - |
| IDPA-Caps (Tsai et al., 2020) | 2.99±0.18 | 6.81±0.31 | 27.01±0.50 | 42.19±0.48 | 26.29±0.37 |
| SR-Caps (Hahn et al., 2019) | 3.12±0.13 | 7.86±0.16 | 28.53±0.28 | 57.86±0.50 | 51.42±0.43 |
| Trans-Caps (ours) | 3.02±0.14 | 6.56±0.16 | 25.17±0.21 | 37.46±0.33 | 15.35±0.29 |

Table 3: The test error rates of various models on familiar and novel SmallNORB viewpoints.

| | | Azimuth | | Elevation | |
|---|---|---|---|---|---|
| | # Params. (K) | Familiar | Novel | Familiar | Novel |
| CNN | 897.2 | 8.42±0.48 | 22.43±1.32 | 7.82±0.63 | 18.97±0.92 |
| DR-Caps | 1406.9 | 8.28±0.50 | 19.33±1.02 | 7.57±0.46 | 17.18±0.88 |
| EM-Caps | 98.1 | **7.25±0.68** | 14.11±0.98 | **6.39±0.81** | 12.73±0.80 |
| VB-Caps | 108.2 | **7.21±0.48** | 11.74±0.73 | 7.12±0.58 | 12.04±0.68 |
| SR-Caps | 989.2 | 7.85±0.61 | 18.47±0.97 | 6.89±0.72 | 16.62±1.29 |
| Trans-Caps (ours) | 103.5 | **7.28±0.53** | **8.34±0.91** | **6.43±0.60** | **9.48±0.77** |

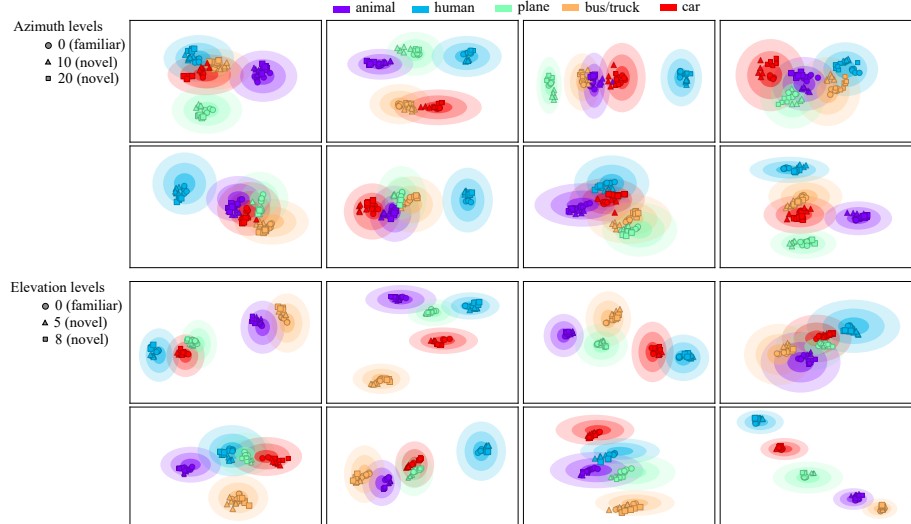

Figure 3: The learned IMoG overlayed with the familiar and novel outputs of the `ClassCaps` layer for the novel azimuth (top) and elevation (bottom) experiments on SmallNORB. Each shade of the Gaussians represents one standard deviation, while each axis represents one of the 16 total poses.

## 4.4 GENERALIZATION TO NOVEL VIEWPOINTS

We evaluate the performance of the proposed CaspsNet architecture on the SmallNORB dataset to demonstrate its generalization strength to novel viewpoints. The SmallNORB dataset is composed of gray-scale $96 \times 96$ images of 5 classes of toys (four-legged animals, human figures, airplanes, trucks, and cars). There are 10 physical instances of each class (5 for train and 5 for test), each of which is pictured at 18 different azimuths, 9 elevations, and 6 lighting conditions (for a total of 48,600 images). The tightly controlled aspects of the SmallNORB dataset make the classification of its images an ideal shape recognition task, and allow us to evaluate the ability of the network to generalize in three-dimensional space. We investigate the ability of the architecture to generalize to both familiar and novel viewpoints. The familiar experiments are performed by training and testing on all viewpoints, while the novel experiments are performed by holding out the most unique azimuths (from 6 to 28) and elevations (from 3 to 8), following what was described in Hinton et al. (2018). We use a single convolution layer backbone with a kernel size of 5 and a stride of 2 to encode the input image into 64 feature maps due to the set size and simplicity of the data. The architecture is described in detail in Table 5 of appendix A.4. To compare performance, we evaluate baseline CapsNet architectures using the same structure and a simple CNN (see Table 11 of appendix A.4).

We report the performance of all architectures in Table 3 for the familiar and novel viewpoint tasks. We note that EM-Caps, VB-Caps, and Trans-Caps significantly outperform the other architectures on both of the familiar viewpoint tasks. In the novel viewpoint tasks, Trans-Caps dramatically outperforms the other architectures, outperforming EM-Caps by $5.77\%$ and $3.25\%$ on the novel azimuth and elevation tasks, respectively. To understand the relationship between the familiar and novel viewpoints in the space of the Gaussians, we plot the learned IMoGs overlayed with the familiar and novel outputs $\boldsymbol{O}_j$ from the `ClassCaps` layer, as shown in Figure 3. We note that the familiar viewpoint outputs (circles) are often grouped adjacent to the center of their respective Gaussians. Novel viewpoint outputs (triangles and squares) tend to cluster with their respective azimuths and elevations, and are therefore typically shifted within the area of their class Gaussians. This implies that the network has learned to represent the input objects in three-dimensional space, allowing it to effectively adapt to novel viewpoints.

## 5 CONCLUSION

In this work, we propose a novel CapsNet architecture named Trans-Caps which employs a non-iterative self-attention routing mechanism to address the computational complexity and numerical instability of CapsNets. The proposed routing mechanism uses a learnable, inducing mixture of Gaussians to estimate the agreement (or attention) between capsules. Our experimental results showed that Trans-Caps can effectively scale up to much larger datasets and outperform the baseline SOTA CNNs and CapsNets in several image recognition tasks with significantly fewer parameters and reduced computational cost. Our observations show that the network is able to effectively construct a three-dimensional understanding of the geometry of objects, which is indicative of properly encoded instantiation parameters.

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

## A    APPENDIX

### A.1    DATA PRE-PROCESSING AND AUGMENTATION

In all our experiments, we consider the same data augmentation for all networks (i.e. CNNs and CapsNets with different routing mechanisms). For SVHN, CIFAR-10, and CIFAR-100 training, we first pad pixels with four zeros and randomly crop the image to $32 \times 32$. For Tiny-ImageNet, images are rotated randomly by up to 20 degrees. For ImageNet, all training images are resized to $256 \times 256$, then randomly cropped to yield an input training patch of $224 \times 224$. For all datasets except SVHN, training images are also horizontally flipped with probability 0.5. Finally, all images are normalized to zero mean and unit standard deviation. During evaluation, we do not perform data augmentation for all models.

For the SmallNORB dataset and following the training protocol of Hinton et al. (2018), all images are resized to $48 \times 48$. During training, each image is randomly cropped to yield an input training patch of $32 \times 32$. Then, we horizontally flip the image with probability 0.5, and randomly jitter the brightness and contrast of the images. During inference, we crop the center of the test image to yield a $32 \times 32$ patch.

### A.2    EFFECT OF THE INDUCING MIXTURE OF GAUSSIANS (IMoGs)

Here we evaluate the role of the IMoG on performance by comparing the performance of our architecture (i.e. IMoG with log-likelihood attention) to a CapsNet which adapts the inducing points (IPs) with dot-product attention from Set Transformers. We see that the increased complexity of the task is positively correlated with the performance gains from the IMoG.

Table 4: Comparison of the mean test classification error rates ($\pm$std.) of the proposed Trans-Caps equipped with IMoGs with log-likelihood attention to IPs with dot-product attention.

|  | SVHN | CIFAR-10 | CIFAR-100 |
|---|---|---|---|
| IMoGs with log-likelihood attention | 3.02$\pm$0.14 | 6.56$\pm$0.16 | 25.17$\pm$0.21 |
| IPs with dot-product attention | 3.08$\pm$0.18 | 7.13$\pm$0.21 | 28.75$\pm$0.32 |

### A.3    EFFECT OF THE NUMBER OF ATTENTION HEADS

We evaluate the effect of the number of attention heads on the performance of the proposed architecture (see Table 5). Our results show that varying the number of heads may alter performance due to the changes in the capsule type's encoded features.

Table 5: The mean test classification error rates ($\pm$std.) of the proposed Trans-Caps with SAR as a function of the number of attention heads.

| # of heads | SVHN | CIFAR-10 | CIFAR-100 |
|---|---|---|---|
| 1 | **3.02$\pm$0.14** | 6.68$\pm$0.21 | 25.32$\pm$0.24 |
| 2 | 3.13$\pm$0.13 | **6.56$\pm$0.16** | **25.17$\pm$0.21** |
| 4 | 3.23$\pm$0.17 | 6.69$\pm$0.19 | 25.42$\pm$0.25 |
| 8 | 3.12$\pm$0.14 | 6.76$\pm$0.22 | 25.38$\pm$0.25 |
| 16 | 3.15$\pm$0.13 | 6.76$\pm$0.20 | 25.50$\pm$0.31 |

## A.4 MODEL CONFIGURATIONS

The following tables depict the architectures and parameter setups for the trained models discussed in the paper. Note that ConvCaps and FCCaps layers represent the convolutional capsule layers and fully-connected capsule layers, respectively.

Table 6: Architecture of the proposed CapsNet with SAR for SVHN and CIFAR-10.

| Name | Operation | Output size |
|---|---|---|
| Backbone ($A = 64$) | ResNet-20 (input_dim=3, output_dim=64) | $8 \times 8 \times 64$ |
| PrimaryCaps ($B = 32$) | $3 \times 3$ Conv, input_dim=64, output_dim=512, stride=1, padding=1 + reshape to 32, $4 \times 4$-dim capsules | $8 \times 8 \times 32 \times 4 \times 4$ |
| ConvCaps ($C = 32$) | $3 \times 3$ ConvCaps SAR to 32, $4 \times 4$-dim capsules, stride=2, padding=1 | $4 \times 4 \times 32 \times 4 \times 4$ |
| ClassCaps | FCCaps SAR to 10, $4 \times 4$-dim. capsules | $10 \times 4 \times 4$ |
| Classifier | input_dim=16, output_dim=1, linear | $10 \times 1$ |

Table 7: Architecture of the proposed CapsNet with SAR for CIFAR-100.

| Name | Operation | Output size |
|---|---|---|
| Backbone ($A = 128$) | ResNet-32 (input_dim=3, output_dim=128) | $8 \times 8 \times 128$ |
| PrimaryCaps ($B = 32$) | $3 \times 3$ Conv, input_dim=128, output_dim=512, stride=1, padding=1 + reshape to 32, $4 \times 4$-dim capsules | $8 \times 8 \times 32 \times 4 \times 4$ |
| ConvCaps1 ($C = 32$) | $3 \times 3$ ConvCaps SAR to 32, $4 \times 4$-dim capsules, stride=2, padding=1 | $4 \times 4 \times 32 \times 4 \times 4$ |
| ConvCaps2 ($D = 32$) | $4 \times 4$ ConvCaps SAR to 32, $4 \times 4$-dim capsules, stride=2, padding=1 | $32 \times 4 \times 4$ |
| ClassCaps | FCCaps SAR to 100, $4 \times 4$-dim. capsules | $100 \times 4 \times 4$ |
| Classifier | input_dim=16, output_dim=1, linear | $100 \times 1$ |

Table 8: Architecture of the proposed CapsNet with SAR for Tiny-ImageNet.

| Name | Operation | Output size |
|---|---|---|
| Backbone ($A = 128$) | ResNet-32 (input_dim=3, output_dim=128) | $16 \times 16 \times 128$ |
| PrimaryCaps ($B = 32$) | $3 \times 3$ Conv, input_dim=128, output_dim=512, stride=1, padding=1 + reshape to 32, $4 \times 4$-dim capsules | $16 \times 16 \times 32 \times 4 \times 4$ |
| ConvCaps1 ($C = 32$) | $3 \times 3$ ConvCaps SAR to 32, $4 \times 4$-dim capsules, stride=2, padding=1 | $8 \times 8 \times 32 \times 4 \times 4$ |
| ConvCaps2 ($D = 64$) | $8 \times 8$ ConvCaps SAR to 64, $4 \times 4$-dim capsules, stride=2, padding=1 | $64 \times 4 \times 4$ |
| ClassCaps | FCCaps SAR to 200, $4 \times 4$-dim. capsules | $200 \times 4 \times 4$ |
| Classifier | input_dim=16, output_dim=1, linear | $200 \times 1$ |

Table 9: Architecture of the proposed CapsNet with SAR for ImageNet.

| Name | Operation | Output size |
|---|---|---|
| Backbone ($A = 1024$) | ResNet-50 (input_dim=3, output_dim=1024) | $14 \times 14 \times 1024$ |
| PrimaryCaps ($B = 64$) | $1 \times 1$ Conv, input_dim=1024, output_dim=1024, stride=1, padding=0 + reshape to 64, $4 \times 4$-dim capsules | $14 \times 14 \times 64 \times 4 \times 4$ |
| ConvCaps1 ($C = 128$) | $3 \times 3$ ConvCaps SAR to 128, $4 \times 4$-dim capsules, stride=2, padding=1 | $7 \times 7 \times 128 \times 4 \times 4$ |
| ConvCaps2 ($D = 128$) | $3 \times 3$ ConvCaps SAR to 128, $4 \times 4$-dim capsules, stride=1, padding=0 | $5 \times 5 \times 128 \times 4 \times 4$ |
| ClassCaps | FCCaps SAR to 1000, $4 \times 4$-dim. capsules | $1000 \times 4 \times 4$ |
| Classifier | input_dim=16, output_dim=1, linear | $1000 \times 1$ |

Table 10: Architecture of the proposed CapsNet with SAR for SmallNORB.

| Name | Operation | Output size |
|---|---|---|
| Backbone ($A = 64$) | $5 \times 5$ Conv, input_dim=1, output_dim=64, stride=2, padding=1 | $15 \times 15 \times 64$ |
| PrimaryCaps ($B = 8$) | $1 \times 1$ Conv, input_dim=64, output_dim=128, stride=1, padding=0 + reshape to 8, $4 \times 4$-dim capsules | $15 \times 15 \times 8 \times 4 \times 4$ |
| ConvCaps1 ($C = 16$) | $3 \times 3$ ConvCaps SAR to 16, $4 \times 4$-dim capsules, stride=2, padding=0 | $7 \times 7 \times 16 \times 4 \times 4$ |
| ConvCaps2 ($D = 16$) | $3 \times 3$ ConvCaps SAR to 16, $4 \times 4$-dim. capsules, stride=1, padding=0 | $5 \times 5 \times 16 \times 4 \times 4$ |
| ClassCaps | FCCaps SAR to 5, $4 \times 4$-dim. capsules | $5 \times 4 \times 4$ |
| Classifier | input_dim=16, output_dim=1, linear | $5 \times 1$ |

Table 11: Architecture of the baseline CNN used in SmallNORB experiments.

| Name | Operation | Output size |
|---|---|---|
| Conv1 | $5 \times 5$ Conv, input_dim=1, output_dim=64, stride=2, padding=1 + BN + ReLU | $15 \times 15 \times 64$ |
| Conv2 | $1 \times 1$ Conv, input_dim=64, output_dim=128, stride=1, padding=0 + BN + ReLU | $15 \times 15 \times 128$ |
| Conv3 | $3 \times 3$ Conv, input_dim=128, output_dim=256, stride=2, padding=0 + BN + ReLU | $7 \times 7 \times 256$ |
| Conv4 | $3 \times 3$ Conv, input_dim=256, output_dim=256, stride=1, padding=0 + BN + ReLU | $5 \times 5 \times 256$ |
| AvgPool | $5 \times 5$ global average pooling + flatten | $256$ |
| Classifier | input_dim=256, output_dim=5, linear | $5$ |

