# OpenReview forum: "Trans-Caps: Transformer Capsule Networks with Self-attention Routing"
_ICLR.cc/2021/Conference — Reject_

### Official Review · AnonReviewer1 · 2020-10-20
**Bringing attention into capsules breaks unique object-part relations.**

**Rating:** 4
**Confidence:** 4

**Review:**

The paper proposes to swap the typical routing mechanisms in capsules for a more standard attention mechanism. The attention mechanism is based on computing similarity scores using gaussians instead of dot-products . The authors show that this leads to better downstream performance of more natural tasks while preserving robustness to viewpoint changes, one of the main strengths of capsules.

My main concern is that the proposed solution is not a good fit for the general framework of capsules. One of the key ideas behind capsuels is that objects are made up of parts which are uniquely assigned to exactly one object. This assumption is broken in this paper (if I understood the paper correctly). That is, the attention mechanism doesn't enforce any kind of competition between objects for individual parts. Hence, this model should be thought of as a stacked set transformer instead. The results indicate that this is not necessary for downstream performance (it actually helps) and the models still remain somewhat robust to view point changes on synthetic data (NORB).

== Detailed Comments and Questions ==

Where is the "self" in self-attention? IIUC there is no attention between individual states of one layer, not even indirectly. There is only attention between some learnable "object" states (inducing points) to their individual parts.

Why is it necessary to model the likelihood of a part state in this model, when all we care about is the similarity between an objects inducing vector and a part representation. Why not using just simple dot-product attention to compute the similarities? The paper says it is used to "encode the second order interactions among points". What does that mean? Actually, the log probability used as similarity score can be written as a dot-product (ignoring the norms of the vectors) + some bias over j which doesn't matter when computing the softmax later on.

I don't understand the difference to self-routing capsules (Hahn et al.). The paper says they use stationary routing weights to specific locations, but I don't see how this approach differs from that? The routing weight in Hahn et al. basically computes dot products between inducing points (the rows in the learnable routing weight matrix) and the output of the part capsule which are used as similarity scores. The only difference here is that gaussians are used to model similarity, which, as I explained above is pretty much the same as a dot-product. Something that's different is the use of multiple heads and using a softmax over parts instead of over objects, which brings me to my biggest concern.

I am also not sure that this architecture fits well into the Capsules framework. CapsNets make the assumption that each part belongs to a single object. That's also the reason why typically there needs to be some iterative procedure to compute probable assignments. Here, however, parts can be part of multiple objects, because the softmax is taken over the parts and there are multiple heads. This kind of defeats the purpose of capsules.

Ultimately, the architecture is basically a stacked version of the set transformer with some potentially interesting deviations, and it should be presented as such with the necessary ablations. I think it is interesting to see that "competition" between objects for the individual parts is not necessary to achieve similar or better performance to capsnets, to achieve good performance on down stream tasks.

---

> ### Author Response · Authors · 2020-11-23
> **Author Responses to Reviewer#1**
>
> Thank you for your valuable and detailed review. We have clarified all the concerns in our updated draft.
>
> 1. __Where is the "self" in self-attention?__
> - Thank you for your questions. The set-transformer essentially computes the similarity between input and inducing points. This is a form of self-attention. It allows the network to construct elementwise relationships between the inputs across multiple layers (i.e. as opposed to doing so in one operation as with traditional self-attention mechanisms). This is similar to what is being described in Big Bird by Manzil, Guru, et al. (2020) in their Global Attention operation. They connect all elements of their input sequence to a global node, which then bias a given element to a higher-level output element, allowing for the same relationships to be determined between the two elements that would be exhaustively constructed in a self-attention operation. We adopted a similar idea in our self-attention routing (SAR) mechanism to compute the similarity among the votes cast from the lower-level part capsules. This allowed us to reduce the quadratic time complexity of the element-wise self-attention operation to linear time complexity, while still allowing us to construct key relationships between discriminative input votes which we then use to generate the instantiation parameters of the higher-level objects.
>
> 2. __The log probability used as similarity score can be written as a dot-product__
> - In our model, weights (\Phi) and standard deviations (\Sigma) of the Gaussians are trained discriminatively with backpropagation; this allows the network to weight the distance of an input with respect to a set of Gaussians in different regions of the solution space (compared to the uniformly distributed spaces with a dot-product operation with inducing points). This allows for the Gaussian distributions to account for the wide range of variability we observe in the instantiation parameters of an object, which may vary in complexity and range. We also included an ablation study in the appendix A.2 to highlight the advantages of our method when compared to the simple dot-product equivalent.
>
> 3. __The difference to self-routing capsules (Hahn et al.)__
> - Self-routing capsules employ a similarity score computed across the pose of the part capsules. This approach computes similarity with the lower-level capsule coordinate frames, and as a result, requires a total of i*j (where i is the number of input capsules and j is the number of output capsules) inducing points (if we assume that the rows of the routing matrix are inducing points). Our approach instead computes similarity in the higher-level capsule space. We first generate our votes by mapping the lower level instantiation parameters into the higher-level capsules coordinate frames. We then utilize j Gaussians to compute the similarity scores using the log-likelihood attention mechanism we describe in the paper. This strategy makes it easier for our routing mechanism to draw conclusions from the various input sources while using a significantly smaller number of induced points.
>
> 4. __Architecture does not fit well into the Capsule frameworks__
> - Our routing mechanism is inspired by the recent talk by Geoffrey Hinton at AAAI-20 (posted at https://player.vimeo.com/video/390347111) and their latest Stacked Capsule Autoencoder paper (Kosiorek et al. 2019) which favor whole-part (as opposed to part-whole) relationships. If we attempt to construct an object (or an object part) from a lower-level part with fewer degrees of freedom, then we expect the direct predictions to be error-prone. For example, we cannot infer the pose of a constellation from the location of a single star (we instead must know the pose of many stars). The dynamic routing mechanism proposed in Sabour et al. (2017) is a bottom-up approach which degrades the part capsule votes according to their similarity to the objects’ predicted pose to finally get a cleaner prediction of the objects’ pose after a few “iterations”. In contrast, our attention-based routing mechanism is a top-down approach where the softmax operation is inverted to enforce competition between the votes cast by the lower-level capsules of the same type when generating the attention scores. It is worth noting that these capsules share the same transformation matrix and are trained to encode the pose of the same part perceived at different spatial locations. This allows us to gate the meaningful information being passed from the child capsules to the objects while effectively suppressing the flow of irrelevant votes from lower-level capsules at non-descriptive spatial regions.
>
> We are happy to answer any other questions you may have.

---

> > ### Comment · AnonReviewer1 · 2020-11-23
> > **Attention between inducing points and inputs != self-attention**
> >
> > I don't understand the reasoning behind comparing attention between inducing points and some input as self-attention.
> >
> > I also don't understand the connection to big bird, which IIUC, actually project the global information back to the input.
> >
> > In this paper, the attention is never related back to the input itself, hence I don't see the "self" in self-attention here. It is just object-to-part attention, not part-to-part attention. After applying SAR, there is no notion of the parts anymore, only the objects. You essentially apply an attention based pooling operation with multiple inducing points on inputs of each layer, IIUC.

---

> > > ### Author Response · Authors · 2020-11-25
> > > **Response to:  Attention between inducing points and inputs != self-attention**
> > >
> > > Thank you for deepening the conversation; it is important to note that the global attention mechanism described in Big Bird (which was introduced in the Longformer) does NOT project the information back into the input. The global attention mechanism uses global tokens (i.e. a [CLS] token in QA global attention), which attend to all tokens in an output sequence while all input tokens attend to it. Let us now think of the keys generated from our capsule votes as input tokens and the IMoGs as tokens with global attention. When we compute our similarity coefficients, the IMoGs are effectively selecting the input tokens that are most descriptive of the object of interest. The selected input tokens, therefore, __communicate with each other indirectly through the IMoGs__ in our object-to-part SAR mechanism. This leads to your next comment; our intention was to form object-to-part (not part-to-object) relationships. It is difficult to understand the context of a part (i.e. what is this a “part” of?) without having some prior knowledge about the possible objects it can be associated with. This is especially true when we are attempting to construct these relationships in complex domains; while it is easy to understand the relationship between parts and objects in MNIST through part-to-object routing, it is significantly more challenging to do so in natural image domains. If we attempt to learn this part-to-object relationship discriminatively through backpropagation, we are introducing a potential point of failure that can limit the performance of the network, as is implied by our comparisons to other CapsNet architectures. This is the critical difference between our work and Hahn et al., which we discuss in the next comment.

---

> > ### Comment · AnonReviewer1 · 2020-11-23
> > **The log probability used as similarity score can be written as a dot-product**
> >
> > When taking the softmax over i in Eq. 4, all the additive components that only rely on j, will be erased, which leaves us:
> >
> > a_j =  softmax_i [  s_{i,j}  ]
> >       =  softmax_i  [ -  (k_{i,j} - \mu_j)^2 / (2*\sigma^2_j)  ]
> >       =  softmax_i  [ (2 <k_{i,j}, \mu_j> -  k_{i,j}^2)   / (2*\sigma^2_j)  ]
> >
> > That's, if I understand correctly basically the same as a dot-product but somewhat normalized by the norm of k_{i,j} and sigma. Since sigma_j is the same for all i, it basically just serves as a learnable temperature. Given this derivation I would be curious what would happen if you were to simply do dot-product attention with a learnable temperature per capsule and maybe normalize k using l2 norm as well.
> >
> > In any case, please correct me if that is not almost identical to dot-product attention with some minor tweaks (which are interesting). Maybe I am missing something here.

---

> > > ### Author Response · Authors · 2020-11-25
> > > **Response to: The log probability used as similarity score can be written as a dot-product**
> > >
> > > We agree with the reviewer that this equation is approximately similar to the dot-product with the exception of our k_{ij} and \Sigma_j terms. By subtracting by k_{ij} in the numerator, we allow the network to bias the routing coefficient depending on the input capsule i. More importantly, by normalizing by sigma, we are weighing the distance of the input to each Gaussian depending on the respective “spread” of the output capsule. This allows our network to account for the complexity in the appearance of each object. We can expect more complex instantiation parameters to take on a larger range of values, which we encode directly into our IMoGs. Another critical characteristic of our architecture is that, due to the inherent properties of IMoGs, we are able to enforce competition between the object capsules (because \sum phi_j = 1), which would not be possible using IPs with temperature. These changes grant an advantage to our architecture as shown in appendix A.2.

---

> > > > ### Comment · AnonReviewer1 · 2020-11-25
> > > > **RE: there is (still) no competition between objects and sigma does not make this model more flexible.**
> > > >
> > > > IIUC then there is still no competition because as I have explained before because \phi_j has no part in computing the a_ij (see my derivation above).
> > > >
> > > > Normalizing by sigma is something that's also not necessarily more expressive because a model without sigma_j could simply learn \mu_j with different norms. That would have an identical effect.

---

> > ### Comment · AnonReviewer1 · 2020-11-23
> > **The difference to self-routing capsules (Hahn et al.)**
> >
> > "We then utilize j Gaussians to compute the similarity scores using the log-likelihood attention mechanism we describe in the paper."
> >
> > --> Isn't this as well a i*j operation similar to Hahn et al.s W^route projection? Note that W^route_j in Hahn et al., are essentially your \mu_j stacked together. See my comment below as I still believe you are doing some variation of dot-product attention similar to Hahn et al, with a minor caveat.
> >
> > Besides, what's this "likelihood" connection that's mentioned? Just writing down some notion of probability and then using softmax on the log_prob is a pretty hand-wavy connection imo.

---

> > > ### Author Response · Authors · 2020-11-25
> > > **Response to: The difference to self-routing capsules (Hahn et al.)**
> > >
> > > The most important difference between our routing mechanism and Self-routing is that we are constructing the routing weights in the "object" domain. In Self-routing, the routing coefficients are constructed using the __pose of the part capsules__ (equation 3 in Hahn et al. 2019), meaning that information from the object capsule coordinate frames are encoded into these coefficients through backpropagation alone (i.e. part-to-object relationships are constructed discriminatively in a single pass). This is problematic; single-pass part-to-object routing mechanisms cannot make direct use of information encoded in the object coordinate space. As we have discussed in our previous response, part-to-object routing is a very challenging problem to solve; depending on backpropagation alone to establish relationships between the part and object coordinate frames limits the potential of Self-routing in more complex domains (e.g. CIFAR-100, ImageNet, etc.). By generating our votes prior to our routing coefficients, we encode object-to-part relationships discriminatively in a single pass, which greatly improves the ability of the architecture to adapt to new viewpoints, for example.
> > >
> > > We have used the term “likelihood” because of the similarity between our approach and the Maximum Likelihood of the mixture of Gaussians (i.e. the likelihood of the K_{ij} data points with respect to the mixture of Gaussians, as shown in equation 3). The key difference is that we are training a neural network to learn parameters \Mu, \Phi, and \Sigma, thus allowing the routing mechanism to treat each vote differently through the output attention coefficients.

---

> > > > ### Comment · AnonReviewer1 · 2020-11-25
> > > > **RE: There is no MoG after the softmax over the inputs.**
> > > >
> > > > I understand that Hahn et al apply the softmax over the objects, which I believe is the main difference. And I agree that direct part-to-object routing is a problem, hence conceptually simpler attention mechanisms as here and in the set transformer are just easier to train. Therefore, I am advocating and pointing out that this model doesn't fit well to the original ideas of capsules. I don't agree that making capsules almost identical to attention and not acknowledging this fact is good for the paper, and therefore misleading. The whole notion of MoG allows for modeling more complex object-part relationships compared to attention is simply not true. Please have a closer look at the derivation of the routing weights in my other comment. There is no significant difference to standard attention that would justify the complexity that's used to derive this model. I would be much happier with the paper if it would drop the whole notion of MoGs and capsules, and shows how a fairly simple stacked set transformer does much better on the typical tasks used to evaluate capsules.
> > > >
> > > >
> > > > Note that \Phi is not trained in your model as I have explained in another answer, because when taking the softmax over the i (the inputs), \phi_j disappears, and so does the whole notion of MoG. Btw., this should be easy to verify by looking at the gradients of the phi when training your model. If your model is implemented as described in the paper they should always have a zero gradient.

---

> > ### Comment · AnonReviewer1 · 2020-11-23
> > **Architecture does not fit well into the Capsule frameworks**
> >
> > I believe that the formulation where competition is taken over the "parts" is essentially just attention (as you note as well). For my taste this way of formulating the architecture just fits better into the attention literature. To me it seems like attention and the set transformer is "rediscovered" in this paper. The only novelty I see is that there is a little twist to the dot-product attention that allows for a learnable temperature per capsule and normalizes the inputs before applying the dot-product.
> >
> > Please note that the softmax over the inputs in Eq.4 of the paper makes the whole notion of gaussians and likelihoods a bit meaningless, because mixtures of gaussians compete between themselves and represent a form of clustering of the input. That's not happening here.

---

> > > ### Author Response · Authors · 2020-11-25
> > > **Response to: Architecture does not fit well into the Capsule frameworks**
> > >
> > > It is important to note that the competition is taken over both the parts (using the softmax among the part capsules to filter out outlier votes from irrelevant spatial locations) and the objects (using the IMoGs with learnable parameters), not over the parts alone. This is in line with our previous responses about object-to-part routing and is discussed in more detail in the previous replies. We also should note the similarity between capsule routing and the self-attention mechanism initially used in Transformers; in both cases, we are trying to bias certain input elements based on the relationships between those elements. This similarity has been widely recognized by the community and has resulted in various novel routings architectures, such as STAR-Caps (Ahmed et al. 2019), Self Routing Capsules (Hahn et al. 2019), and SCAE (Kosiorek et al. 2019). Please see the other responses regarding the role of the IMoGs in our architecture and why they improve performance.

---

> > > > ### Comment · AnonReviewer1 · 2020-11-25
> > > > **RE: This model is (still) not doing MoG**
> > > >
> > > > Please read my other comments. There is no MoG in this model given that you apply the softmax over the inputs. The whole notion of it just falls apart. Also read my other comment about how I think this model is too close to self-attention and the set transformer to be put into the capsules camp.

---

### Official Review · AnonReviewer2 · 2020-10-28
**Novel non-iterative scalable Capsule Network which improves viewpoint generalization too**

**Rating:** 7
**Confidence:** 3

**Review:**

Authors propose a novel Capsule connection which is inspired by set transformer and induced points. They introduce log-likelihood based attention based on Gaussians centered at trainable fixed queries. They calculate the routing factors (attention) using the probability of a key projection of votes under trainable gaussians. Then they add the values weighted by the routing factor to the fixed mean. Their choice of modifying the mean based on the input rather than replacing it is quite interesting. It resonates with the concept of momentum as well. I am curious how much it affects the stability and convergence of their technique.
Afterwards they linearly transform and add a skip connection + relu to get the output capsule parameters.
In this work the activation probability of the capsules are not calculated alongside the pose matrices. For the sake of classification (which needs the activation probabilities) they have an extra fully connected layer + cross entropy.
Their method surprisingly can generalize to new viewpoints, backed by experiments on smallNorb azimuth and elevation generalization much better than the CNN and previous Capsule Networks.
Also by removing the iterative routing and replacing it with trainable parameters they are able to achieve competitive results on Cifar10-Cifar100-tiny imagenet and imageNet.

Pros:
The paper is very well written and easy to follow. They provide a convincing set of experiments on reasonable datasets. Their method is novel and intuitive.
Cons:
Lack of ablation study to show the importance of their novel method IMoG. One baseline is just the attention used by set transformer. Essentially what is the effect of having a Gaussian (standard deviation). Is it necessary to modify the mean vs replacing it.
It would be more convincing to add attention based vision models (bello et al 2019) to the tables for cifar10-cifar100.



------------------------Post Author Response


Thank you for adding the ablation study and the attention based models. I enjoyed reading your work and it has answered some of the questions we wanted to explore in Capsule Networks.

---

> ### Author Response · Authors · 2020-11-22
> **Author Responses to Reviewer#2**
>
> Thank you for your valuable and detailed review. We have clarified all the concerns in our updated draft.
>
> 1. __Effect of the IMoGs__
> - Thank you for the suggestion. We have included an ablation study to investigate the role of the IMoG. Specifically, we include a comparison to inducing points with dot-product attention (proposed in set transformers by Lee et al. (2019)). We observed that IMoG improved the performance, while only slightly increasing the model complexity (additional parameters Gaussian weights and standard deviations). This result indicates that the network is able to make use of the weighted similarity (or distance) metric that we construct using our learned standard deviation, which is designed to account for the variation present in the instantiation parameters of an object.
>
> 2. __Add attention-based vision models to the results table__
> - Thank you for the suggestion. We have added two non-Capsule baseline methods, namely the Squeeze and Excitation networks (SE-ResNet) method by Hu et al. (2018) and attention-augmented convolution networks (AA-ResNet) by Bello et al. (2019), to our results.
>
> We are happy to answer any other questions you may have.

---

### Official Review · AnonReviewer3 · 2020-10-30
**The paper proposes a self-attention (learning based) routing method that improve the performance over the baselines but there are a couple of unjustified points that need clarification.**

**Rating:** 6
**Confidence:** 4

**Review:**

The paper proposes to use the self-attention to find the agreement among the capsules of consecutive layers of a capsule network instead of iterative routing procedure. To reduce the computational and memory complexity of self-transformer capsules of each layer are considered as a set and follow set-transformer and apply inducing mixture of Gaussian distributions to compute the agreement among capsules of layer L and L+1.
The proposed method have been evaluated on multiple dataset including large scale datasets like ImageNet and improved the performance compared to the baselines (except for  ImageNet)
pros:
1- The idea is interesting and the paper is well written and easy to follow.
2- The proposed method seems to be scalable to the large datasets.
3- It achieves significant improvement compared to the baselines on the novel view point of SmallNorb.

cons(clarification)
1- There are related papers that are missing in the comparison section for instance
-Capsule routing via variational Bayes.
2- Why is the performance the proposed method on the ImageNet is even less than the baseline Resnet50.
3- It seems that increasing the number of attention head negatively affect the performance, what is your justification for that?
4- Results in the table 3 is counterintuitive. The proposed method has a significant improvement compare to the baseline on the novel view point of SmallNorb while it is on par with other method in familiar viewpoint.

Post Rebuttal

Thanks for the author(s)' responses. The rebuttal addressed some of my questions. I have a couple of  suggestions :
1- Your proposed capsule network is not the first one that is applicable on large scale datasets like ImageNet, there are other capsule networks that are applicable on real world scenarios and also ImageNet dataset with improvements over the baseline
- Dual Directed Capsule Network for Very Low Resolution Image Recognition (ICCV 2019)
-  Subspace Capsule Network (AAAI 2020)
please refer to them and also give intuitions about why your proposed Trans-Caps is not performing well on ImageNet. The intuition and analysis is valuable to the community.

2- To support the generalizability claim of Trans-Caps, I highly recommend reporting results on Multi-MNIST and also affNIST. Specially when you train the model on MNIST and test it on these two datasets.

---

> ### Author Response · Authors · 2020-11-22
> **Author Responses to Reviewer#3**
>
> Thank you for your valuable and detailed review. We have clarified all the concerns in our updated draft.
>
> 1. __Add more related baselines to the results section__
> - Thank you for your suggestion, we have added VB-Caps by Ribeiro (2020) and IDPA-Caps by Tsai (2020) to our comparisons. We perform 5-fold cross-validation and report our results on a separate test dataset in all experiments, as explained in the paper.
>
> 2. __Performance on the ImageNet dataset__
> - We did not mean to claim that the proposed capsule networks outperform SOTA deep networks on all datasets. Our goal is, however, to show the first capsule network architecture with stable training and competitive performances on real-world image datasets. Our results are significant because training existing capsule networks to work with ImageNet and other complex datasets are still extremely challenging.
>
> 3. __Effect of the number of attention heads__
> - When we increase the number of attention heads, we effectively devote a fixed number of feature maps (i.e. pose matrix dimensions) to each of the attention operations. We are therefore reducing the number of features that are used in our SAR mechanism per attention head. If we do not account for the reduced number of features available to each attention head, we change the way that our mechanism encodes the information and may risk harming performance due to the altered discriminative ability of each attention head. For example, we may observe improved performance when utilizing 32 features across 2 attention heads as opposed to 8 features across 8 attention heads; this is task-dependent and may change. We have included this explanation to the corresponding appendix section.
>
> 4. __Counterintuitive results in Table 3__
> - In the familiar viewpoints task, we can assume that the similar performance across the evaluated networks is due to the simplicity of the task. This means that the performance across the evaluated architectures is similar simply because they are all capable of solving the simple 5-class recognition problem; this makes sense since the input training data is representative of the validation data, which differs only slightly from what the model has already seen. Under novel viewpoints, the task requires models to have stronger generalizations. It is well-known that capsule networks were designed to generalize better under viewpoint changes. The fact that our results exceeded SOTA performance in this task confirmed that our networks can construct a better 3D representation, allowing it to solve the problem when observing an object from an entirely novel viewpoint.
>
> We are happy to answer any other questions you may have.

---

### Official Review · AnonReviewer4 · 2020-10-30
**Review: Trans-Caps: Transformer Capsule Networks with Self-attention Routing**

**Rating:** 6
**Confidence:** 3

**Review:**

The submission details a novel technique to learn the routing in capsule networks for image classification tasks. Connecting capsules in such architectures typically requires iterative approaches which are computationally expensive. The main idea in this submission is to leverage a non-iterative attention mechanism to learn this routing and thus decreases computational cost. Furthermore, the experiments indicate that the proposed architecture leads to higher accuracy on a number of image classification tasks and datasets.

Overall, the submission is well written and easy to follow. The discussion of related work is thorough and to my best knowledge appears to be complete.

---
## Pros:

   * The paper clearly presents the proposed method, and it is easy to follow.
   * The results show significant improvements over the baselines, especially on the Tiny-ImageNet dataset and the SmallNORB dataset in the case of novel poses.
   * The proposed learning algorithm does not require expensive iteration and thus allows for better scaling.
   * The good results across several datasets suggest that the proposed model can be an important contribution to the field.

---
## Cons:

   * The authors discussion of related work mentions several papers that combine CapsNets with an attention mechanisms to address the issue of routing.
   * However, the authors only provide a comparison to Hahn et al. (2019). The authors state that other methods are memory intensive and require knowledge of the number of concurrent iterations as an additional hyperparameter, which is probably why they do not compare to these methods.
   * Although an important property, it would be beneficial to provide a comparison to other attention-based CapsNets.
   * Furthermore, it would be interesting to compare to state-of-the-art image classification methods that are not based on the capsules concept. Adding these additional experiments would help readers to have a better overview of the task and where the proposed approach stands in the broader context of classification approaches.
   * In other areas of deep-learning and representation learning in particular, use of GMMs in the architecture, e.g., in latent space, (as opposed to usage as a probabilistic output model) can lead to code-book collapse. I.e., it is hard to balance during pure back prop learning how to pick the correct gaussian to sample from and to shape it  (or in other words: should one update the parameters of a mixture component or penalize the selection of that component if the action leads to a high loss in the forward pass). It would be interesting to see more details on this aspect of the architecture and training. I.e. how was the number of mixtures selected, how does this influence performance? etc.

---
Summary:
Overall, there is much to like about this submission but some questions remain (see cons). I'm looking forward to the authors' response.

---

> ### Author Response · Authors · 2020-11-22
> **Author Responses to Reviewer#4**
>
> Thank you for your valuable and detailed review. We have clarified all the concerns in our updated draft.
>
> 1. __provide a comparison to other attention-based CapsNets__
> - Thank you for the suggestion. We have added VB-Caps by Ribeiro (2020) and IDPA-Caps by Tsai (2020) to our comparisons. We perform 5-fold cross-validation and report our results on a separate test dataset in all experiments, as explained in the paper.
>
> 2. __compare to state-of-the-art image classification methods__
> - Thank you for the suggestion. We have added two popular approaches, namely Squeeze and Excitation networks (SE-ResNet) by Hu et al. (2018) and attention-augmented convolution networks (AA-ResNet) by Bello et al. (2019).  These methods are a more up to date representation of the state-of-the-art performance in non-capsule, attention-based CNN performance.
>
> 3. __Number of IMoGs and collapse of the Gaussian components__
> - The number of IMoGs is a tunable hyperparameter (similar to the number of filters in a convolutional layer) that corresponds to the number of object capsules that we are encoding in a given layer. We did not experience code-book collapse in our experiments. However, if this problem occurs, we can use multiple random initializations of GMMs (including the number of clusters, Gaussian means, and standard deviations) and pick the best model using the standard cross-validation procedure.
>
> We are happy to answer any other questions you may have.

---

### Author Response · Authors · 2020-11-25
**Update**

Dear Reviewers and AC,

We sincerely appreciate all the reviews. They give positive and high-quality comments on our paper with a lot of constructive feedback and concerns which help clarifying many important points. We have updated our draft to incorporate the insightful suggestions of the reviewers. The major changes are as follows:

- We have added VB-Caps by Ribeiro (2020) and IDPA-Caps by Tsai (2020) as additional baseline CapsNets to our comparisons.

- We have added two popular approaches, namely Squeeze and Excitation networks (SE-ResNet) by Hu et al. (2018) and attention-augmented convolution networks (AA-ResNet) by Bello et al. (2019) as state-of-the-art, non-capsule, attention-based CNNs.


- We have included an ablation study (Appendix A.2) to investigate the role of the proposed IMoGs by comparing its performance to inducing points with dot-product attention (proposed in set transformers by Lee et al. (2019).


Thank you all for the valuable suggestions.
Please let us know if you have additional questions.

Thank you,

Authors

---

### Decision · Program_Chairs · 2021-01-07
**Final Decision**

**Decision:**

Reject

**Comment:**

The paper proposes a new variant of capsule networks, where iterative routing is replaced by an attention-based procedure inspired by Induced Set Attention from Set Transformers. The method is competitive on several classification benchmarks and improves generalization to unseen views on SmallNORB.

The reviewers note that the method is presented well (R2, R3, R4), is more scalable than other capsules variants (R3, R4), and the results are good (R1, R2, R3, R4). However, the reviewers also point out missing relevant baselines (R2, R3, R4), limited amount of generalization experiments (R3), and issues with the positioning of the method and the details of the formulation (R1). In particular, R1 did a very thorough job at reading the paper and discussing with the authors.  The issue with missing baselines has been satisfactorily addressed in the updated version of the paper.

Considering all this feedback and after reading the paper myself, I would summarize the pros and cons of the paper as follows.

Pros:
1. Good presentation
2. The method is more scalable than prior capsule-based models
3. Competitive results on several small- to mid-scale classification datasets
4. Good results on viewpoint generalization on SmallNORB

Cons:
1. Classification results on all datasets are worse than non-capsules models (SE-ResNet, AA-ResNet). I could not find a discussion of this fact either in the paper, or in the authors’ responses. Given this fact, superior generalization (or some other nice properties) would be a potential advantage of the proposed model. Which leads to the next point.
2. Generalization results on SmallNORB are encouraging, but it is just a single dataset. If these results are key to showing the benefit of the method (as argued in the previous point), it is crucial to demonstrate this generalization in more settings, e.g. at least on MultiMNIST and AffNIST, as suggested by R3.
3. Scalability of the method is only studied in limited detail (I do appreciate Figure 2). The best indication in the direction of scalability is that the model can be trained on ImageNet (which is great), but it performs worse than the ResNet-50 used as a backbone and it is not explained why (even after one of the reviewers asked about it) and how expensive computationally the model is.
4. I share the concerns of R1 regarding the use of the term “MoG”. It is a mathematical term, so one would expect mathematical precision when using it.
  4a. It is unclear how the mixing probabilities \phi are learned (IIUC they get no gradient, as described by R1) and if they are in some way actually learned, it is unclear how it is guaranteed that they sum to one.
  4b. MoG usually comes with the standard procedure of fitting it to data (EM), which IIUC the authors are not following here. This should be clearly explained.
5. A relatively more minor concern: again, as pointed out by R1, the use of “self-” in “self-attention” does not seem accurate. Self-attention assumes inputs to the attention procedure attend to themselves in some sense. As one consequence, the output sequence has the same length as the input sequence. ISAB from Set Transformer can be seen as a factorized version of self-attention where first inducing points attend to the inputs and then the inputs attend to the inducing points, so the output of the whole block is still the same length as the input. But in the proposed model this second step of going back to the inputs is absent and the length of the output sequence is generally different from the length of the input sequence.

Note:
I partially share the doubts R1 raised on the positioning of the method as “capsules” as opposed to “attention”, but I believe it is not the authors’ fault that the definition of what capsules are is historically vague and that this term has been used in many different ways in the past. I would strongly recommend to discuss this point in the updated version of the paper and I hope the capsules community manages to get more clarity on what exactly capsules are. But I do not count this point as a weakness here.

Based on all this evidence, I recommend rejection at this point. The paper has its merit, but it has unfortunate gaps both on the experimental and the presentation sides, as listed above. Some of these have been mentioned during the discussion phase, but the authors have not quite addressed them. There is no mechanism to ensure these are fixed in the final version, so resubmission to a different venue is the only option.